# Temperature-Independent Lifetime and Thermometer Operated in a Biological Window of Upconverting NaErF_4_ Nanocrystals

**DOI:** 10.3390/nano10010024

**Published:** 2019-12-20

**Authors:** Kailei Lu, Yingxin Yi, Li Xu, Xianhao Sun, Lu Liu, Hanyang Li

**Affiliations:** 1College of Physics and Optoelectronic Engineering, Harbin Engineering University, Harbin 150001, China; kaileilu@163.com (K.L.);; 2Key Lab of In-fiber Integrated Optics, Ministry Education of China, Harbin Engineering University, Harbin 150001, China; 3School of Economics and Management Engineering, Harbin Engineering University, Harbin 150001, China

**Keywords:** upconversion, temperature-independent lifetime, NaErF_4_, thermometer

## Abstract

Lifetime of lanthanide luminescence basically decreases with increasing the ambient temperature. In this work, we developed NaErF_4_ core–shell nanocrystals with compensation of the lifetime variation with temperature. Upconversion lifetime of various emissions remains substantially unchanged as increasing the ambient temperature, upon 980/1530 nm excitation. The concentrated dopants, leading to extremely strong interactions between them, are responsible for the unique temperature-independent lifetime. Besides, upconversion mechanisms of NaErF_4_ core-only and core–shell nanocrystals under 980 and 1530 nm excitations were comparatively investigated. On the basis of luminescent ratiometric method, we demonstrated the optical thermometry using non-thermally coupled ^4^F_9/2_ and ^4^I_9/2_ emissions upon 1530 nm excitation, favoring the temperature monitoring in vivo due to both excitation and emissions fall in the biological window. The formed NaErF_4_ core–shell nanocrystals with ultra-small particle size, highly efficient upconversion luminescence, unique temperature-independent lifetimes, and thermometry operated in a biological window, are versatile in applications such as anti-counterfeiting, time-domain manipulation, and biological thermal probes.

## 1. Introduction

Lanthanide (Ln) heavily doped nanocrystals (NCs), with intense energy harvest and, thus, both potentials of efficient upconversion (UC) luminescence as well as light-to-heat conversion, are promising for a broad range of applications spanning background-free biolabeling and biosensing, light-triggered drug delivery, multimodal phototherapy, 3D display, solar cell, and light-emitting diodes [1,2,3,4,5,6,7]. However, Ln ions have abundant ladder-like energy levels, and cross-relaxation (CR) between ions and energy transfer (ET) to the surface quenchers become more sensitive with increasing the doping level; thus, the UC intensity of activating ions (such as Er^3+^, Tm^3+^, Ho^3+^, and Nd^3+^) heavily doped upconversion nanocrystals (UCNCs) has been strictly limited [8,9]. Over the past decade, researchers have developed various strategies to suppress the concentration quenching effect, mainly through coating an inert shell [10,11,12], increasing excitation power density [13], choosing host NCs with a large unit cell size [14], and engineering the distribution of dopants [15].

Among the above-mentioned strategies, the core–shell structure is one of the most effective ways to suppress the high-doping concentration quenching in UCNCs. Very recently, Johnson et al. [16] and Zuo et al. [17] have shown that the most intense luminescence is achieved when the host lattice is fully occupied by activators (100 mol.% doping), after inert layer passivation. Since then, Chen et al. [18] and Shang et al. [19] achieved enhanced UC luminescence in Er^3+^ heavily doped NCs via energy condensation through combined effects of mediated transient energy trapping and inert-shell coating. Although highly efficient UC luminescence of high-doping NCs has been reported in recent years, UC mechanisms have not been discussed in detail, and especially some unique optical properties aroused by high concentration doping remain undiscovered [20,21].

Conventional applications of UCNCs are generally based on the emission intensity and color. Recently, time-domain of UC is increasingly concerned, as it provides new dimension to exploit the UC emission, and thus multiplexing applications such as anti-counterfeiting, optical coding, and multichannel labeling are possible [22,23]. However, it is well known that the lifetime of Ln luminescence is sensitive to temperature, which decreases with heating, resulting in difficulty of manipulating lifetime under various ambient temperatures [24]. Thus, achieving temperature-independent lifetime of UCNCs is highly desired.

In addition, remote, sensitive, and accurate temperature monitoring is of vital importance for in vivo applications such as drug delivery and photothermal therapy [25]. The fluorescence intensity ratio (FIR), usually based on luminescent intensities from two thermally coupled levels of lanthanide ions, is the most popular method adopted for UC thermometer [26,27]. In principle, the populations of two thermally coupled levels follow the Boltzmann distribution, which requires the energy gap below 2000 cm^−1^, and thus sensitivity of the conventional FIR thermometer is restrained, due to it is proportional to the energy gap. Alternatively, using non-thermally coupled energy levels for FIR thermometer can greatly increase the sensitivity [28]. Further, choosing proper non-thermally coupled UC bands within the biological window not only improves the sensitivity, but also deepens the in vivo operation. Finally, high-doping core–shell UCNCs possessing highly efficient UC luminescent can improve sensing accuracy due to the high signal to noise ratio. Based on above discussions, Ln heavily doped core–shell UCNCs are promising thermal probes for in vivo applications.

In this paper, we investigated the UC luminescence behaviors of ultra-small NaErF_4_ core-only and core–shell NCs excited by 980 and 1530 nm laser. Effects of inert shell coating were discussed on the basis of structural and spectral measurements, thus enabling the analysis of the UC mechanisms. Anomalous variation of UC lifetimes in Er^3+^ heavy-doping NCs at various ambient temperatures were, for the first time, found and discussed. On the basis of the bright UC emissions, the FIR sensing behaviors of the thermal coupling levels (^2^H_11/2_ and ^4^S_3/2_) and non-thermally coupled levels (^4^S_3/2_/^4^F_9/2_, and ^4^I_9/2_) of Er^3+^ are comparatively explored.

## 2. Experiment Section

### 2.1. Synthesis of NaErF_4_ Core-Only NCs

The ultra-small NaErF_4_ core-only NCs were prepared via a slightly modified literature procedure [29], 1 M ErCl_3_, 0.5 M NaOH and NH_4_F methanol solutions were prepared firstly. Then, 1 mL of ErCl_3_ methanol solution, 8 mL of oleic acid (OA), and 15 mL of 1-octadecene (ODE) were added to a 50 mL three-necked flask. The mixture was heated at 180 °C with stir for 45 min before cooling down to 50 °C. Subsequently, 5 mL of NaOH and 8 mL of NH_4_F methanol solution were added into the mixture and stirred for another 45 min. After the reaction mixture was heated at 110 °C for 15 min to remove the residual methanol and water, the solution was quickly heated to 280 °C and kept for 1.25 h before cooling down to room temperature. The as-prepared NCs were precipitated by addition of ethanol, collected by centrifugation at 8000 rpm for 5 min, and washed with ethanol and methanol for several times. The final product was dispersed in 10 mL cyclohexane. Argon gas was adopted throughout the entire experiment to protect the reaction.

### 2.2. Synthesis of NaErF_4_@NaGdF_4_ Core–shell NCs

The as-prepared NaErF_4_ core-only NCs were used as seeds for shell modification. In a typical experiment, the shell solution was first prepared by mixing GdCl_3_ methanol solution (1 M, 0.5 mL), OA (5 mL), and ODE (7.5 mL) in a 50 mL three-necked flask. The resulting mixture was heated at 180 °C for 45 min before cooling down to 50 °C. 2.5 mL of NaOH and 4 mL of NH_4_F methanol solution and 5 mL of NaErF_4_ seed solution were then added and stirred for 45 min. After heating at 110 °C for 15 min to remove the residual methanol and water, the reaction mixture was quickly heated and kept at 280 °C for 1 h. The resulting core–shell NCs were washed and dispersed in cyclohexane following the above mentioned route. Argon gas was used throughout the entire experiment to protect the reaction.

### 2.3. Characterization and Spectral Measurements

The crystal structures of the sample was identified by powder X-ray diffraction (XRD, Smartlab, Rigaku Corp., Akishima, Japan) manufacturer, city, state abbreviation if US or Canada, country under various temperatures, with a resolution of 0.03°/step from 15 to 70°. The particle morphologies were recorded on a transmission electron microscopy (TEM, Tecnai G2, FEI Co., Ltd., Hillsboro, OH, USA). Room temperature UC luminescence measurements were performed by irradiating samples via a variable-power 980 or 1530 nm diode laser (LWIRL980-5W and LWIRL1530-1W, Laserwave Optoelectronics Technology Co., Ltd., Beijing, China). The emissions of core-only and core–shell *β*-NaErF_4_ NCs, dispersed into cyclohexane with identical concentration, were recorded by a spectrometer (FLMS03177, Ocean Optics Co., Ltd., Orlando, FL, USA). For time resolved luminescence measurements, the continuous wave lasers were modulated into square-wave output by a signal generator (VC2002, Victor Instruments Co., Ltd., Shenzhen, China), and the decay profiles were recorded by a photomultiplier tube (CR131, Zolix Instruments Co., Ltd., Beijing, China), connected to an oscilloscope (TBS1102, Tektronix Co., Ltd., Shanghai, China). To obtain the temperature dependent UC spectra and lifetimes, the powder form samples were pressed into small disks, which were further placed onto an electric heating plate (IKAC-MAG HP4, JKI Co., Ltd., Shanghai, China), with the temperature resolution of 0.1 K.

## 3. Results and Discussion

### 3.1. Characterizations

Figure 1a depicted the temperature-dependent XRD peaks of NaErF_4_, which are consistent with the standard data of hexagonal NaErF_4_ (PDF# 27-0689). No additional peaks appear, indicating the as-prepared sample is highly pure *β*-NaErF_4_. The crystal phase remains unchanged with increasing the ambient temperature up to 600 K, whereas the refraction peaks gradually blue-shift (Figure 1b), indicating the lattice expansion. The core-only NCs are well mono-disperse due to the good surface modification of OA. Spherical NaErF_4_ NCs are highly uniform in size, with an average diameter of 11.6 nm (Figure 1c). After epitaxial growth of pure NaGdF_4_ shell (~1.6 nm) on the core seeds, the mean diameter of core–shell NCs increases to ~14.9 nm, evidencing the successful coating of the inert shell (Figure 1d). 

### 3.2. UC luminescence Properties

To illustrate the effect of inert coating, UC spectra of NaErF_4_ core-only and core–shell NCs were recorded (Figure 2). Upon NIR (980 or 1530 nm) laser excitation, the NCs exhibit green, red, and 800 nm UC emission bands, respectively corresponding to the transitions of ^2^H_11/2_/^4^S_3/2_→^4^I_15/2_, ^4^F_9/2_→^4^I_15/2_, and ^4^I_9/2_→^4^I_15/2_. The luminescence intensity of core-only NCs is extremely weak owing to strong concentration quenching, leading to no emission peak detected at 980 nm excitation. As more “dark ions” are activated through ~4-fold absorption intensity of Er^3+^ at 1.5 μm than that at 1 μm [21], slight emissions appear when pumped by 1530 nm laser.

In stark contrast, core–shell NCs yield highly efficient UC emissions (980 or 1530 nm excitation). The enhanced factor of overall intensity between core and core–shell (*I*_core–shell_/*I*_core_) NCs obtained by 1530 nm excitation goes up to ~1100, and it reaches ~8 for 1530 and 980 nm (*I*_1530_/*I*_980_) in core–shell NCs. The significant enhancement can be mainly attributed to the effective suppression of surface quenching, as core–shell structure extends the distance between luminescence centers and surface vibration modes. It is noteworthy that the relative intensity of NIR to visible emission in core NCs is stronger than that in their core–shell counterparts, due to visible luminescence originated from higher energy levels is more susceptible to the surface defects compared to the lower transition of ^4^I_9/2_→^4^I_15/2_.

Further, pumping power dependent UC intensity of NaErF_4_ core-only and core–shell NCs under NIR excitation were investigated. The relationship between the UC intensity and the excitation power, *I*_UP_ ∝ *P^n^*_NIR_, can be deduced by a simplified rate equation model [30], where *I*_UP_, *P*_NIR_, and *n* refer to the UC intensity, excitation power, and the number of NIR photons involved in the UC processes, respectively.

As plotted in Figure 3, the slopes obtained by linearly fitting are 1.2–1.5 (Figure 3a) and 2.1–2.2 (Figure 3b), indicating 2-photon UC process in core–shell NCs stimulated at 980 nm and 3-photon process in core-only NCs at 1530 nm, respectively [31,32]. However, slope values lower than 2, obtained in core–shell NCs upon 1530 nm excitation (Figure 3c), obviously differ from its core-only counterpart. These decreased slopes can be attributed to the effects of inert passivation. Specifically, owing to core–shell structure effectively suppresses surface quenching, upward transitions dominate downward relaxations in Er^3+^ intermediate levels of ^4^I_13/2_ and ^4^I_9/2_, and thus the slopes in the core–shell NCs decreases [33,34]. The detailed UC pathways of Er^3+^ excited by 980 or 1530 nm were illustrated in Figure 3d. Due to the inert passivation eliminating the high-energy surface vibration modes, red emitting level ^4^F_9/2_ should be less populated by nonradiative decay from green level ^4^S_3/2_. As a consequence, CR1 and CR2 are, respectively, responsible for the populations of red emission upon 980 and 1530 nm excitation [21].

### 3.3. Anomalous Variation of Lifetime Versus Temperature

Time-resolved luminescent intensities were measured to shed more light on the UC behaviors. As shown in Figure 4a, the decay curves of laser and core-only sample are almost identical, indicating a rapid decay of core-only NCs (≤40 μs). After inert coating, the lifetime prolongs to 120 and 170 μs. The rapid decay of core-only NCs can be attributed to the strong surface quenching induced by ultra-high concentration (100 mol.%). When surface quenchers are isolated from Er^3+^ by an inert shell, the detrimental ETs to the surface quenchers are eliminated. Consequently, lifetime increases in the core–shell NCs, consistent with the highly enhanced luminescent intensity. It is noteworthy that the measured lifetime of core–shell NCs is smaller than that obtained in NaErF_4_@NaLuF_4_ (10 nm shell thickness) [16], which can stem from the thinner shell used in the present work (~1.5 nm).

Also, the measured lifetimes are obviously smaller than expected. On the basis of the J-O calculation of 80 mol.% Er^3+^ doped NaGdF_4_, which is similar to NaErF_4_, transition probabilities of Er^3+^ fall within 10^1^~10^3^ s^–1^ (^4^S_3/2_:240 s^–1^, ^4^F_9/2_:640 s^–1^, and ^4^I_9/2_:70 s^–1^) [20].

From another side, the non-radiative decay rates follows
(1)$$W_{\mathrm{nr}}=W_{0}\exp(-\alpha \Delta E)$$
where *W*_0_ = 1 × 10^8^ s^−1^ and *α* = 5 × 10^−3^ cm are constants, estimated by using LaF_3_ crystal [35], which is similar to NaErF_4_ (hexagonal structure and cut-off phonon energy of ~350 cm^−1^); Δ*E* is the energy gap given in cm^−1^, within 2000~3000 cm^−1^ for Er^3+^ UC emitting levels (^4^S_3/2_:3117 cm^−1^, ^4^F_9/2_:2885 cm^−1^, and ^4^I_9/2_:2228 cm^−1^) [36]. According to Equation (1), the decay rates of ^4^S_3/2_, ^4^F_9/2_, and ^4^I_9/2_ levels are 20, 50, and 1450 s^−1^, respectively. Combining the radiative transition and phonon-assistant decays and omitting the ETs and surface quenching, UC lifetimes of NaErF_4_ core–shell NCs should be within 0.7~4 ms. However, the measured UC lifetimes are all below 0.3 ms, indicating other processes dominating the lifetime. We attributed them to the strong interactions between Er^3+^, accelerating the depopulation of excited levels through CR processes, which were not labeled in Figure 3d as there are numerous possibilities.

Besides, it is found that the 980 nm laser excited green and NIR lifetimes are generally larger than that obtained by using 1530 nm laser, whereas lifetimes of red emission are similar (Figure 4a and Table 1). In principle, lifetime of an energy level depends on several processes including radiative transitions, non-radiative decays (from its upper level and to its lower level), and ETs involved. The spontaneous transition probabilities of lanthanide ions are the intrinsic natures of the luminescent centers and surrounding crystal field; multiphonon-assistant decay rates are related to the energy gap and lattice vibration modes; ET rates are decided by the spectral overlap of donor and acceptor as well as their spatial distance. Therefore, for a given luminescent materials with fixed composition, above three parameters should be independent to the incident wavelength. As a consequence, the lifetime variation aroused by incident wavelength can stem from the different UC pathways. As shown in Figure 3d, green emitting levels of ^4^S_3/2_ and NIR level of ^4^I_9/2_ are not directly populated by the incident 980 nm photons, whereas 1530 nm photons directly feed these levels. This results in the prolonged lifetimes excited by 980 nm laser, as electrons populating ^4^S_3/2_ and ^4^I_9/2_ levels undergo additional decays from their upper levels. As for the red emission, ^4^F_9/2_ level is fed by CR processes for both 980 and 1530 nm excitations, and extremely strong interactions of Er^3+^ lead to similar lifetime upon various excitations.

Most importantly, lifetime of NaErF_4_ core–shell NCs remains substantially unchanged with increasing the ambient temperature (Figure 4b and Table 1). It is well known that lanthanide luminescence is susceptible to ambient temperature. In general, lifetime decreases with increasing the temperature (see Figure 4b, 1/3 reduction in conventional 2Er/18Yb NCs), due to the stronger non-radiative decay processes associated with phonon vibrations [37]. 

This temperature-independent variation might be the consequence of the lattice expansion with heating, which can be proven by the blue-shift of the XRD peaks with increasing temperature [Figure 1b] and is in good agreement with the observations elsewhere [38,39,40]. As mentioned above, CR processes are dominant for Er^3+^ UC lifetimes. Similarly, there should be considerable energy migrations between excited Er^3+^ in high-doping situation. The energy migrations occur through dipolar-dipolar interactions, where the efficiency is inversely proportional to the 6th power of the average donor-acceptor distance *r*. Thus, for the energy migration process through the consecutive ETs of *n* pairs of Er^3+^, the efficiency is proportion to *r*^−6*n*^ [41]. As a consequence, Er^3+^ luminescence process is strongly influenced by the energy migration distance. Increased temperature causes the lattice to expand, resulting in a larger transfer distance *r* and a longer transfer time, and finally increase the Er^3+^ lifetimes. Herein, the prolonged lifetime aroused by lattice expansion compensates the shortened lifetime caused by thermal quenching, leading to the temperature-independent lifetimes observed. This phenomenon relies on high-doping concentration, thus does not occur in traditional 2Er/18Yb sample (Figure 4b), and is also consistent with the observation that the prolonged lifetimes of Yb^3+^ (higher concentration) are more evident than that of Eu^3+^ (lower concentration) in NaGdF_4_ with increasing temperature [39].

### 3.4. Temperature Sensing 

Typical UC spectra of NaErF_4_ core–shell NCs under 1530 nm excitation show the quenched intensities with raising temperature, and the relative intensity also varies with temperature (Figure 5). Therefore, based on the FIR technique, temperature sensing behavior of NaErF_4_ core–shell NCs under 1530 nm excitation was investigated. The FIR from thermally coupled levels (^2^H_11/2_ and ^4^S_3/2_) can be expressed as follows [42],
(2)$$FIR=\frac{I_{2}}{I_{1}}=C\exp(\frac{-\Delta E}{kT})+A$$
and FIR from non-thermally coupled levels (^4^S_3/2_, ^4^F_9/2_ and ^4^I_9/2_) can be fitted by a polynomial function [43]
(3)$$FIR=\frac{I_{2}}{I_{1}}=B_{0}+B_{1}T+B_{2}T^{2}$$
where *I*_i_ is the integrated intensity of the emission band; *k* is the Boltzmann constant; *T* is the absolute temperature; *A*, *B*_i_, and *C* are the constants need to be determined.

For thermometers, the sensitivities (absolute sensitivity *S*_a_ and relative sensitivity *S*_r_) are critical for evaluating their sensing performance, which can be determined according to the expressions:(4)$$S_{a}=\frac{d\left(FIR\right)}{dT}$$
(5)$$S_{r}=\frac{d\left(FIR\right)}{dT}\frac{1}{FIR}$$

Figure 6a,b depict the FIR values as a function of temperature within the range of 303–593 K, by using thermally coupled and non-thermally coupled levels. The experimental data can be well fitted according to Equation (2) or (3), and the corresponding sensitivities are also shown in Figure 6d–e. The sensitivity obtained by using non-thermally coupled levels of ^4^S_3/2_ and ^4^I_9/2_, both for *S*_a_ or *S*_r_, are higher than that obtained by using thermally coupled levels, the conventionally used thermally-coupled levels. The maximum *S*_a_ and *S*_r_ reach 0.0149 K^−1^ at 593 K and 1.15% K^−1^ at 303 K, respectively, within the operating temperature range, comparable to results reported recently, as shown in Table 2. 

It is noteworthy that both the excitation wavelength and the Er^3+^ emissions of ^4^F_9/2_ and ^4^I_9/2_ level fall in the so-called biological windows (І-BW spinning 650–950 nm and Ш-BW within 1500–1750 nm [51]), the low-loss wavelength region in biological tissue. Combining this with other fascinating features of the newly developed UCNCs such as the ultra-small particle size (low biotoxicity) and the highly efficient UC emissions (high signal to noise ratio), using red and NIR UC emissions are promising for the in vivo temperature feedback. Well fitted data shown in Figure 6c evidences the feasibility of this biological thermometer. The maximum sensitivities of 0.0024 K^−1^ and 0.59% K^−1^, both obtained at 303 K which is close to the body temperature (Figure 6f), are slightly higher than that from the traditional ^2^H_11/2_/^4^S_3/2_ FIR.

## 4. Conclusions

In summary, the ultra-small hexagonal-NaErF_4_ core-only and core–shell NCs were synthesized by thermal decomposition method. Optical properties, including luminescence intensity, emitting lifetime, and UC mechanisms were investigated. The inert shell significantly enhances the luminescent intensity by a factor of ~1100, upon 1530 nm excitation. The unique temperature-independent lifetimes were obtained, mainly attributed to the balance between lattice expansion (prolong the lifetime) and thermal quenching (shorten the lifetime). Moreover, use of non-thermally coupled levels for FIR thermometry exhibited the maximum absolute and relative sensitivities of 0.0149 K^−1^ and 1.15% K^−1^, respectively, evidently higher than that obtained through traditional thermally coupled FIR. Finally, thermometer on the basis NIR/red FIR, suitable for in vivo temperature feedback, was demonstrated. These meaningful results will deepen the understanding of UC luminescence processes in Er^3+^ heavily doped systems, and also manifest the wide application prospects of NaErF_4_ core–shell UCNCs.

## Figures and Tables

**Figure 1 nanomaterials-10-00024-f001:**
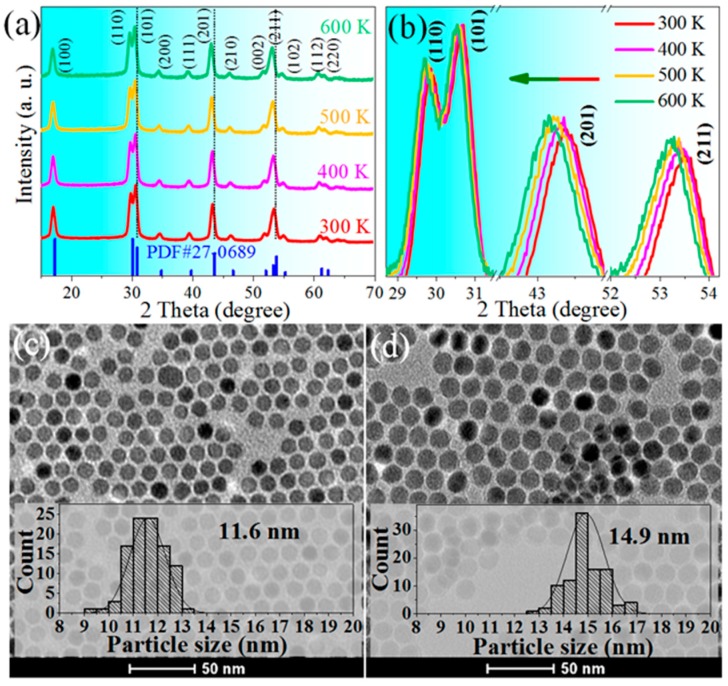
(**a**) Temperature-dependent XRD patterns of the NaErF_4_ core-only NCs. The standard data for *β*-NaErF_4_ (PDF# 27-0689) is shown as a reference; (**b**) the amplified patterns of the three most intensive peaks. TEM images of (**c**) NaErF_4_ core-only NCs with an average size of 11.6 nm and (**d**) NaErF_4_@NaGdF_4_ core–shell NCs of 14.9 nm. Insets are the size distributions obtained from measurements of 100 particles, and scale bars are 50 nm.

**Figure 2 nanomaterials-10-00024-f002:**
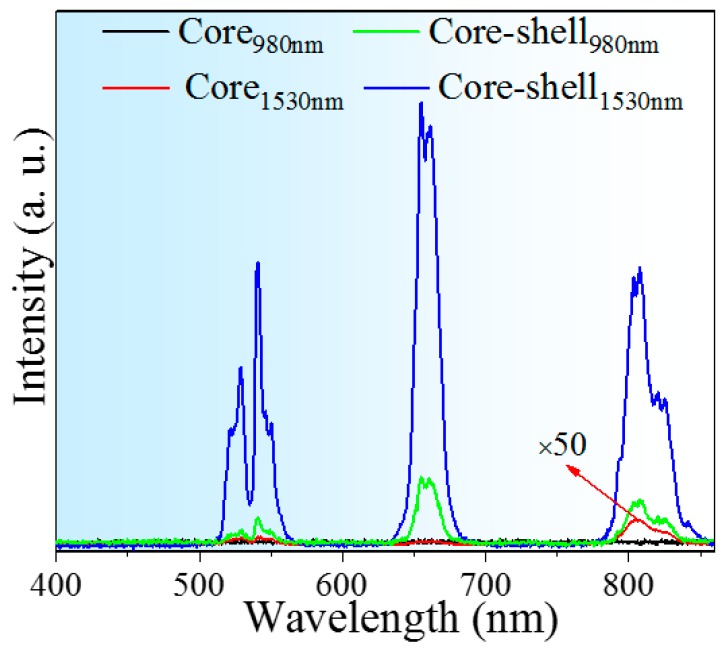
UC spectra of NaErF_4_ core-only and core–shell NCs excited by 980 or 1530 nm laser (10 W/cm^2^), 1530 nm laser induced spectrum of core-only sample is magnified by a factor of 50.

**Figure 3 nanomaterials-10-00024-f003:**
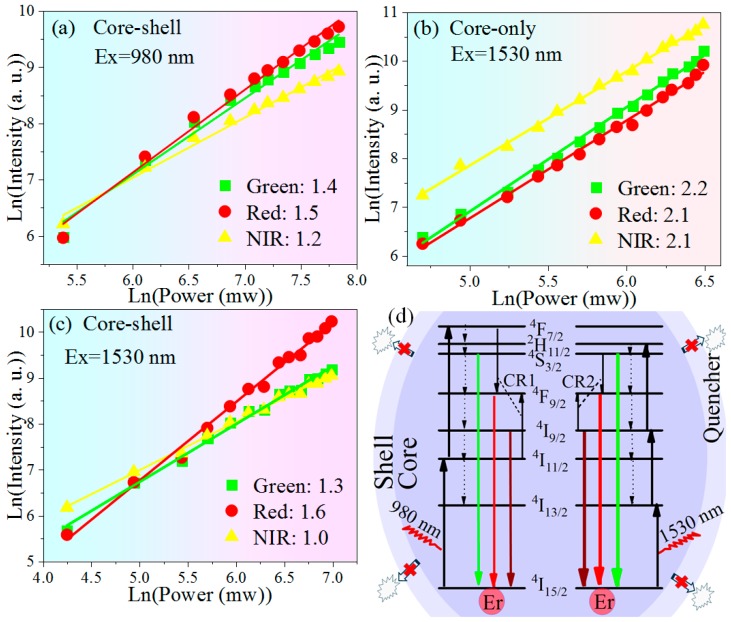
Ln-ln power dependences of Er^3+^ emissions in (**a**) core–shell NCs under 980 nm excitation; (**b**) core-only and (**c**) core–shell NCs excited by 1530 nm laser; (**d**) energy levels and UC pathways of Er^3+^ pumped by NIR (980 or 1530 nm) laser.

**Figure 4 nanomaterials-10-00024-f004:**
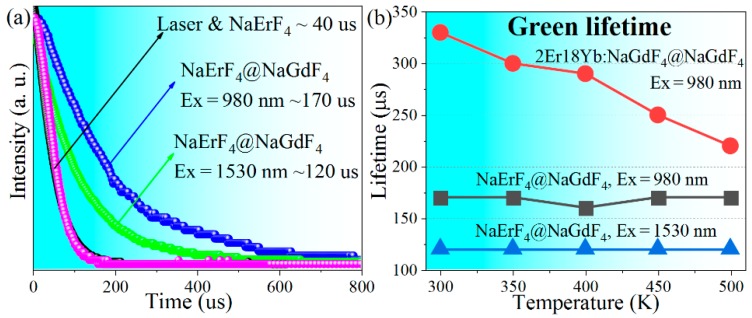
(**a**) Decay profiles of ^4^S_3/2_ emissions of core-only and core–shell NCs under 980 or 1530 nm excitation. 980 nm induced lifetime of core-only NCs is not available due to the extremely weak intensity. Lifetime was estimated by single exponential fitting; (**b**) Er^3+ 4^S_3/2_ lifetimes versus ambient temperature of NaErF_4_ and traditional core–shell NCs.

**Figure 5 nanomaterials-10-00024-f005:**
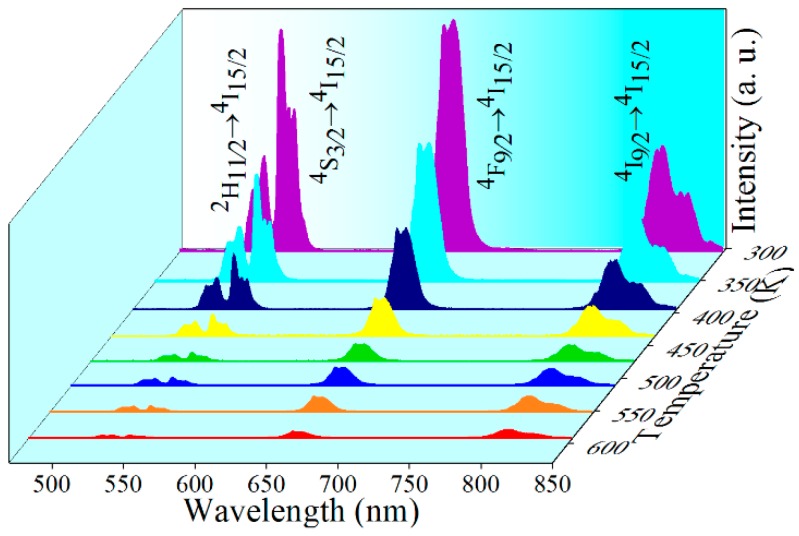
Typical UC emission spectra of NaErF_4_ core–shell NCs in the temperature range 303–539 K, stimulated by 1530 nm laser (20 W/cm^2^).

**Figure 6 nanomaterials-10-00024-f006:**
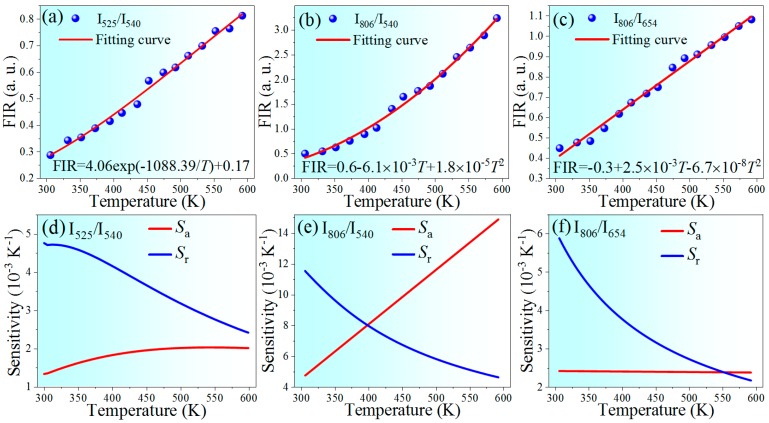
The FIR as a function of temperature based on (**a**) *I*_520_/*I*_540_, (**b**) *I*_806_/*I*_540_, and (**c**) *I*_806_/*I*_654_. Fitting functions are also given; (**d**–**f**) the corresponding relative and absolute sensitivity by using various emission bands.

**Table 1 nanomaterials-10-00024-t001:** Lifetimes of NaErF_4_ core–shell NCs at various temperatures upon 980 or 1530 nm excitation.

Excitation	Initial Level	Temperature-Dependent Lifetime (μs)				
		300 K	350 K	400 K	450 K	500 K
980 nm	^4^F_9/2_	130	130	130	140	140
	^4^I_9/2_	230	230	250	260	260
1530 nm	^4^F_9/2_	140	130	140	140	140
	^4^I_9/2_	170	170	180	180	180

**Table 2 nanomaterials-10-00024-t002:** Comparisons of *S*_a_ and *S*_r_ based on FIR in different materials.

Materials	Transitions	Range (K)	*S*_a-max_ (K^−1^) (maximum)	*S*_r-max_ (% K^−1^)	Ref.
LuVO_4_:Yb/Er	^2^H_11/2_/^4^S_3/2_	303–423	0.0083 (123 K)	1.14	[44]
Y_2_O_3_:Er	^2^H_11/2_/^4^S_3/2_	93–613	0.0044 (427K)	0.98	[45]
Gd_2_O_3_:Yb/Er	^2^H_11/2_/^4^S_3/2_	298–573	0.0078 (528 K)	1.16	[46]
NaYF_4_:Yb/Er@SiO_2_	^2^H_11/2_/^4^S_3/2_	300–900	N.A.	1.02	[47]
NaGdF_4_:Yb/Er	^2^H_11/2_/^4^S_3/2_	303–363	0.0025 (360 K)	1.12	[48]
NaLuF_4_:Yb/Er	^4^D_7/2_/^4^G_9/2_	303–523	0.0052 (303 K)	0.43	[49]
NaYF_4_:Yb/Er	^2^H_11/2_/^4^S_3/2_	303–743	0.0044 (637 K)	0.46	[50]
NaErF_4_@NaGdF_4_	^4^I_9/2_/^4^S_3/2_	303–593	0.0149 (593 K)	1.15	This work
	^2^H_11/2_/^4^S_3/2_		0.0020 (544 K)	0.48	
	^4^I_9/2_/^4^F_9/2_		0.0024 (303 K)	0.59	

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
