# Peer review of "Temperature-Independent Lifetime and Thermometer Operated in a Biological Window of Upconverting NaErF4 Nanocrystals"

_nanomaterials, 2019, doi:10.3390/nano10010024_

Round 1

Reviewer 1 Report

The manuscript is reporting the synthesis of NaErF4 core-shell nanocrystals (NCs) and their optical properties including luminescence intensity, emitting lifetime, and upconversion mechanisms. Introduction of the inert shell over the NaErF4 core significantly enhances the luminescent intensity and showing unique temperature-independent lifetimes unlike the cases of conventional upconverting lanthanide materials. Moreover, based on the NIR/red fluorescence intensity ratio (FIR), thermometric property of the material was demonstrated.

Overall the work and the investigation carried out in this manuscript is important and appealing, but few parts of the manuscript need to support with suitable references and experiments as commented below. Accordingly, the manuscript needs a major revision before publication in this journal.

Comments:

1. “….it is well known that the lifetime of Ln luminescence is sensitive to temperature, which decreases with heating, resulting in difficulty of manipulating lifetime under various ambient temperatures.” This comments by authors should be supported with suitable references.

2. Please mention the procedures for temperature variation dependent luminescence intensity measurement.

3. In page 5, cite the known references to support the comment on slope dependent 2-photon and 3-photon excitation phenomenon.

4. Author concluded that “This temperature-independent variation might be the consequence of the lattice expansion with heating.” Does this phenomenon is observed through XRD?

5. Author have claimed that this materials is suitable for in vivo temperature feedback. It is recommended to demonstrate any in vivo imaging application using this newly developed material.

Author Response

Dear reviewer:

On behalf of my co-authors, we thank you very much for giving us an opportunity to revise our manuscript, we appreciate you very much for your positive and constructive comments and suggestions on our manuscript. We also appreciate your carefulness, conscientiousness, and the broad knowledge on the relevant research field. According to your suggestions and comments, we have made responses and some corrections in our revised manuscript. Our responses to the comments and the content corrections are shown in the submitted PDF, please check it.

Your time and efforts on this paper are highly appreciated and we are looking forward to hearing from you soon again.

Sincerely,

Lu Liu

Reviewer 2 Report

Paper reports on the synthesis of nano-hexagonal-NaErF4 core-only and core-shell NCs via thermal decomposition method. Sythesized NCs are highly uniform and possibly can be used for in vivo applications. Optical properties, including luminescence intensity, emitting lifetime, and UC mechanisms were investigate based on the well-known and understood energy migration processes in Er doped solid state hosts. Also, the passivation effect of the core-shell NCs confirms the expected role of surface recombination processes. Authors claim that NCs with luminescence temperature-independent lifetimes can be obtained by balancing between lattice expansion (prolong the lifetime) and thermal quenching (shorten the lifetime). The concept of temperature measurement by using non-thermally coupled level is not new. There are numerous reports on this topic. Authors shall critically demonstrate the benefit of using the NaErF4 vs. other alternative hosts like https://doi.org/10.1016/j.jallcom.2019.02.047.  

Furthermore, authors shall demonstrate the claimed host lattice expansion experimentally and then correlate it with the claimed lifetime extension.

Some discussion focused on the host phonon spectra distributions, phonons energy and critical comparison of these parameters with other host may help reader to assess the probability of non-radiative and co-operative process in studied UPNCs system. This issue is somehow neglected in manuscript despite its paramount importance for rationality of the proposed model. As the result of that the energy level diagram (Fig,3(d)) is oversimplified and leave much room for speculation.  

In Fig. 1 (b),(c) inserts distribution diagrams are invisible, change the color.

In conclusion, I recommend major revisions prior accepting for publication.

Author Response

Dear reviewer:

Thank you very much for your work on our manuscript. We also appreciate your carefulness, conscientiousness, and the broad knowledge on the relevant research field. According to your suggestions and comments, we have made responses and some corrections in our revised manuscript. Those comments are all valuable and very helpful for revising and improving our paper, as well as the important guiding significance to our researches. Our responses to the comments and the content corrections are shown in the submitted PDF, please check it.

Your time and efforts on this paper are highly appreciated and we are looking forward to hearing from you soon again.

Sincerely,

Lu Liu

Round 2

Reviewer 1 Report

The revised manuscript is suitable for publication.

Reviewer 2 Report

This is revised manuscript. Authors have addressed satisfactorily all reviewer’s concerns and comments. Manuscript is recommended for publication as is.